# Toward Maximizing Assessment Efficiency: A Synthesized Trial-Based Functional Analysis and Competing Stimulus Assessment

**DOI:** 10.3390/bs14050372

**Published:** 2024-04-28

**Authors:** Lesley A. Shawler, Gabriella Castaneda-Velazquez, Grace Lafo

**Affiliations:** School of Psychological and Behavioral Sciences, Southern Illinois University, Carbondale, IL 62901, USA; gabriella.gomez@siu.edu (G.C.-V.);

**Keywords:** synthesized FA, trial-based FA, competing stimuli, competing stimulus assessment, severe SIB

## Abstract

Despite the success of the standard functional analysis (FA), some limitations to conducting an FA in practice include time, resources, ecological relevance, and safety, which have led to the development of procedural adaptations such as trial-based and synthesized FA formats. The purpose of this case study was to identify the function(s) of self-injurious behavior (SIB) for a 3-year-old female with developmental disabilities using a brief trial-based FA with ecologically relevant synthesized contingencies, based on caregiver input, to minimize opportunities for SIB. We identified that positive physical attention likely functioned, at least in part, as a reinforcer for SIB, in less than 42 min. Overall harm to the child as a result of the synthesized trial-based FA was minimal, and the caregiver viewed the modified conditions favorably. We then assessed the role of competing stimuli on SIB rates with the child’s mother and identified two potential items that may compete with attention as a reinforcer for SIB. Our findings highlight the utility and importance of individualized assessment as the first step in the safe treatment of severely challenging behavior.

## 1. Introduction

Families with children with autism spectrum disorder (ASD) report a range of challenges including stress [1], financial burden [2,3], and decreased quality of life [4]. Five to ten percent of individuals with challenging behavior require serious behavior management [3,5]. In particular, self-injurious behavior (SIB) may include significant injury to the individual and may be difficult to treat depending on the function(s) of the behavior [6,7]. Self-injurious behavior (SIB) may present in a variety of different topographies (see [8,9] for reviews) but, ultimately, can result in harm to the individual emitting it. In some cases, SIB may produce significant injury (e.g., tissue damage) to the individual and may be difficult to treat. As such, early intervention for SIB before individuals develop a history of reinforcement for it becomes essential [2]. Research has found that for some children, certain topographies of less harmful SIB can develop into more severe SIB without proper treatment [10]. Moreover, without proper assessment and intervention in early life, SIB could become a chronic issue into adulthood for some individuals [11].

To determine the function(s) of behavior, a functional behavior assessment consisting of indirect and direct measures, followed by an experimental functional analysis (FA; [12]) (for clarity and consistency, we refer to the procedures from [12] as the standard FA), should be conducted. Decades of research on behavioral intervention continue to demonstrate how FA procedures are widely applicable across diverse behaviors, diagnoses, and topographies of challenging behavior [13,14,15]. However, the standard FA methodology has some acknowledged limitations related to time, feasibility, and outcomes [16,17]. First, the standard FA may occasionally produce undifferentiated or ambiguous results, leading to methodological or design format adaptations [12,13,14,15]. For example, ref. [18] conducted a second FA adding escape-from-attention and tangible conditions following elevated responding during the toy play condition of the standard FA. Responding occurred during both conditions, suggesting challenging behavior was maintained partly by an idiosyncratic function of escape-from-attention and access to tangibles. More recently, ref. [19] added an escape-to-tangible test condition (c.f. [20]) in which a highly preferred item was removed before instructing the participant to complete a task following an initial inconclusive standard FA. Contingent upon challenging behavior, the experimenter terminated the task and the participant resumed access to the preferred item for a brief period. 

Second, reviews of the published literature report that FAs are mainly conducted in outpatient settings (50.3%, [15]). Common environmental constraints in clinical practice include the amount of time and resources necessary to complete an FA, which may present an undue risk of harm to the individual or others due to challenging behavior, and difficulty with maintaining tight control over certain variables to isolate possible controlling variables [21]. Third, another challenge relates to the use of an FA with high-risk or dangerous behavior [21]. Some practical issues have been addressed through various FA modifications that have successfully produced clear functional relations between the independent and dependent variables (e.g., synthesized contingencies, trial-based formats). However, in these circumstances, it is important to identify methods that can be completed efficiently to minimize the individual’s prolonged exposure to the evocative scenarios.

Ref. [22] first demonstrated how an FA can be successfully limited to a single opportunity (i.e., trial) which reduces time in assessment and can minimize associated injuries (i.e., trial-based FA). Trial-based FAs have been found to have high correspondence with standard FAs [23] and are also advantageous in settings where it may be difficult to arrange more traditional session-based FAs and promote more ecologically relevant conditions (e.g., establishing operations presented during ongoing school activities). Ref. [22] conducted the first demonstration of a trial-based FA in a classroom setting with two children. All trials were presented in their naturally occurring context throughout the day. A 60-s test trial in which the relevant putative reinforcer contingency was present was followed by a 60-s control trial with unlimited access to the putative reinforcer during naturally occurring classroom activities. For example, during an attention (test) trial, the teacher removed attention but gave the student a brief reprimand immediately after aggression, followed by a 60-s period of undivided attention (control trial). Results showed a clear differentiation of responses between the test and control trials for both children, suggesting that the relevant test condition(s) identified the reinforcer(s) for aggression within only two hours. The trial-based FA provided a preliminary demonstration of an assessment method emphasizing ecological validity and brevity, which would be beneficial in naturalistic or clinical settings. Since its inception, the trial-based FA has been growing in popularity [15] likely due to its brevity and lower exposure to the putative reinforcer(s) while still producing robust outcomes and high validity [23,24].

Combining relevant antecedents and consequences may also be necessary to identify idiosyncratic functions when the standard FA is undifferentiated. Ref. [20] conducted an Interview-Informed Synthesized Contingency Analysis (IISCA) (the authors acknowledge that the IISCA terminology was not created until [25]); however, for consistency, refer to the methodology described by [20] as an FA alternative to address possible time constraints and improve ecological validity in the assessment process. The authors modified the standard FA format in two primary ways. First, the duration of each condition was only 3 min. Second, the number of conditions was decreased to two, namely, a synthesized test condition (establishing operation present) compared to a matched control condition (establishing operation absent) (akin to a pairwise comparison; [21]). Conditions were derived from the initial interview and parent–child interactions and alternated in a multi-element design. Conditions consisted of synthesized antecedents and consequences (e.g., escape from demands to access a preferred item). For example, in [20]’s study, Bob’s test condition included asking Bob to play with specific items in a certain way and being interrupted by the experimenter. When Bob engaged in challenging behavior, he regained access to his preferred way of playing, and the experimenter terminated the interruption. The matched control condition allowed Bob to play with items the way he preferred, with no interruptions from the experimenter. Clear differentiation between the test and control conditions emerged after 15 min, suggesting the synthesized variable present in the test conditions likely maintained Bob’s challenging behavior. Moreover, similar findings were present for all three children, suggesting the variables present in the synthesized test conditions likely maintained challenging behavior. Proponents of the IISCA methodology assert that synthesized contingencies are more representative of the real-world contingencies maintaining challenging behavior that are not captured in the standard FA.

Methodological modifications have been made to enhance the feasibility or utility of the FA in clinical or applied settings [16,21] and when working with individuals with more severe challenging behavior [20]. Moreover, combining FA formats has increased over the last decade (30.4%, [15]). Both previously described FA format alternatives have garnered attention, potentially due to the increased ecological validity, brevity, and utility of the procedures, especially when time and safety are a concern when working with severe challenging behavior (e.g., [15,24,26,27]). Ref. [28] asserts that combining the trial-based FA and IISCA may increase ecological validity, should a synthesized function be implicated, and shorten assessment time as the trial terminates immediately following challenging behavior. For example, ref. [28] compared the standard FA methodology to the trial-based FA [22,29] and a trial-based IISCA for three children with ASD in a clinical setting. The trial-based FA and standard FA were conducted similarly to the procedures described above. The trial-based IISCA procedures included multiple establishing operations and consequences in the test condition (i.e., synthesized) presented as trials in a contrived context. Results for all participants showed differentiation of challenging behavior in the synthesized test conditions compared to the control conditions. When comparing all FA formats, the authors reported generally high correspondence of identified functions across all three formats, supporting the utility of the trial-based IISCA. However, the trial-based IISCA required the least amount of time and was associated with the lowest frequency of challenging behavior. As such, given some of the strengths of both the trial-based FA and IISCA separately, a trial-based IISCA may increase ecological validity and require a shorter assessment time which may decrease risk due to fewer opportunities for challenging behavior [28] and may be more favorable in a clinical context.

## 2. Purpose

The purpose of this case study was threefold. The overarching goal of this study was to identify the function(s) of SIB for a young girl with ASD to develop a function-based treatment. Based on undifferentiated results from the initial standard FA and the severity of the behavior in question, a second purpose was to assess the effects of a more efficient FA format on rates of severe SIB in a clinical setting. To attempt to minimize prolonging the time until treatment and the potential additional exposure to harm, we modified the FA methodology to the single-opportunity format of the trial-based FA using ecologically valid conditions (e.g., [29]). As such, our study extends [28] by exemplifying a successful demonstration of a synthesized trial-based FA to identify the function(s) of severe SIB. A third purpose was to evaluate how certain stimuli compete with the reinforcer for SIB when attention from an adult is unavailable.

## 3. Method

### 3.1. Participant, Setting, Materials

Farah was a 3-year-5-month-old nonvocal Caucasian female diagnosed with ASD, developmental delays, sensory processing disorder, restless leg syndrome, and pica. Farah was referred for the assessment and treatment of SIB and to address deficits in her developmental milestones and communication. Farah engaged in severe SIB, tantrums, and occasional aggression. Farah’s caregivers reported that Farah had engaged in SIB to the point of causing black eyes, bleeding due to scratching, and other bruising on her body from hitting herself. Historically, Farah’s caregiver reported that it was unclear as to the purpose or function(s) of Farah’s SIB as it occurred across a variety of contexts and followed different antecedents. As such, a clear function-based treatment had not been identified, necessitating further evaluation. Due to the potential for risk of injury to Farah, a brief assessment was prioritized. Farah communicated using the Picture Exchange Communication System (PECS; [30]) and gestures. Farah lived with her mother, father, and younger brother who was also suspected of having a developmental disability. Farah’s mother and guardian provided consent for her assessment and treatment. This study was approved by the University Institutional Review Board at Southern Illinois University, Carbondale, #22182.

All sessions occurred in a small, padded therapy room approximately 2 m × 2 m in a university-based clinic. Sessions were conducted for up to 90 min a day, two days a week for seven weeks. Materials included relevant FA items such as different-colored stimuli (to serve as discriminative stimuli for each condition), preferred items, data collection materials (e.g., laptop or paper and pencil), and various task materials (e.g., puzzles, stacking blocks). Moderately and highly preferred items were included as competing stimuli and tangible reinforcers, respectively, during the FAs and were identified via stimulus preference assessments. Task materials for demands were those that Farah had a low probability of completing. Competing stimuli assessment (CSA) items were an iPad and various preferred toys. 

### 3.2. Measurement 

Dependent variables included the percentage of SIB (trial-based FA), the rate of SIB (standard FA and CSA), and the duration of item engagement (CSA). SIB was defined as scratching with one or both hands on her head, arms, hands, legs, feet, or torso; hitting oneself with one or both hands with open or closed fists; pulling her hair with one or both hands; biting her own hands or feet by using an open mouth and clenching the jaw down on her skin; and making forceful contact with her head, an object, a surface, or a person. Item engagement included having one or both hands on an item for more than 2 s with her body or face oriented toward the item, or eyes oriented toward the iPad for more than 2 s. 

Data recorders included two primary undergraduate or graduate clinicians. We recorded data mainly using a pen and paper, except for the standard FA, which was recorded using laptops. For the standard FA, the frequency of each instance of SIB was recorded for each session. The rate was calculated by dividing the total frequency by 5 min (the total session duration) to produce responses per minute. For the trial-based FA, data were recorded as a percentage of trials. If SIB occurred at any point during the trial, a ‘+’ was recorded. If SIB did not occur, a ‘−’ was recorded at the end of the trial. The total number of trials with SIB was divided by the total number of trials and multiplied by 100 to obtain a percentage. For the CSA, the rate of SIB was calculated by recording each instance of SIB and dividing it by the total session duration (i.e., 2 min). The mean rate of SIB was calculated based on the cumulative rate of SIB per item divided by three (presentations). The mean duration of item engagement was calculated based on the cumulative number of seconds with each item, divided by three presentations. 

### 3.3. Interobserver Agreement

For the trial-based FA, a second independent observer viewed videos for 33% of the FA trials. An exact interval-by-interval IOA procedure was used by breaking each 2-min trial into 10-s intervals. IOA was calculated by the number of agreements out of the total number of intervals and multiplying it by 100 to obtain a percentage. An agreement included both data collectors scoring the occurrence or non-occurrence of SIB in each interval. Disagreements were one data collector scoring the occurrence of SIB, while the second observer did not. Overall, there was a mean of 93% agreement during the trial-based FA, suggesting high agreement between data collectors. 

During the clinician- and caregiver-conducted CSAs, total-count IOA was calculated in vivo for SIB and engagement for 50% of the sessions. To calculate the IOA, the smaller count or duration was divided by the larger count or duration and multiplied by 100 for a percentage. The IOA for SIB across all items was a mean of 84% (range, 50–100%). The IOA for engagement across all items was 97.7% (87–100%).

## 4. Procedure—Phase 1

### 4.1. Multiple Stimulus without Replacement (MSWO) Preference Assessment

Clinicians conducted a Multiple Stimulus Without Replacement (MSWO) preference assessment based on the procedures by [31] to identify preferred items for the FA. The MSWO was conducted 2 times (12 trials) during one session. Items presented included bubbles, a light-up spinner toy, a sensory string, a glitter water toy, a pin toy, and a book. However, a preference hierarchy was undiscernible with the MSWO, so clinicians conducted a paired-choice preference assessment.

### 4.2. Paired-Choice Preference Assessment

A paired-choice preference assessment was conducted based on the procedures from [32]. This assessment was conducted in 30 trials such that each item was paired together twice during one session. Items assessed, in order of Farah’s preference, included bubbles, a pin toy, a sensory string, a glitter water toy, a light-up spinner toy, and a book.

### 4.3. Standard Functional Analysis

Clinicians gathered indirect assessment information from Farah’s caregivers using the Functional Analysis Screening Tool (FAST) [33] and the Questions about Behavioral Function (QABF) [34] measures. The results did not reveal clear potential maintaining variables, indicating a need for further investigation. A structured descriptive assessment [35] was then conducted with Farah and her caregiver to observe naturalistic contexts and consequences for SIB informed by results from the FAST and QABF including demand, tangible, and alone situations. Farah engaged in the most SIB during the tangible condition of the structured descriptive assessment. Next, we conducted a standard FA based on the procedures by [12] with the inclusion of a tangible condition [36] based on indirect information and direct assessment observations with the caregiver. The standard FA included three series and consisted of 5 conditions: attention, demand, alone, tangible, and toy play (which served as the control), which were 5 min each.

### 4.4. Synthesized Trial-Based Functional Analysis

Three series of conditions in the standard FA produced no differentiation between the test and control conditions, and there was some responding during the control condition. Rather than continuing to proceed with more series of the standard FA, based on additional discussions with and observations of Farah and her caregiver, clinicians conducted a synthesized trial-based FA as described by [28]. Given that Farah emitted SIB across all test and control conditions in the standard FA, we decided to maintain similar tests of the establishing operations in the modified FA format. However, Farah’s caregiver described, via informal discussion, various contexts and contingencies that typically occurred surrounding SIB at home. In most cases, the caregiver described antecedents and consequences that were provided or removed in more synthesized contexts as opposed to in isolation. As such, we attempted to modify the current conditions and consequences to be more representative of contexts at home by including common stimuli and using a synthesized condition format. For example, we modified the type of attention delivered from the standard reprimand (e.g., “don’t hit yourself”) to physical attention (e.g., picking her up and soothing her), contingent upon SIB after a demand was placed or an item was removed. This change was largely due to anecdotal observations of a preferred attention type for Farah, as well as caregiver discussions as to what occurred at home following SIB during various activities. We also modified the tangible item from a highly preferred item (bubbles), identified in the preference assessment, to her bottle of milk based on caregiver discussion. Taken together, this information contributed to the development of synthesized conditions conducted in a trial format to maintain brevity and ecological validity [37].

Moreover, given the time already spent on assessment during the standard FA (75 min), we also prioritized identifying an effective but efficient assessment. As such, the trial-based FA was selected given its brevity in identifying the potential function(s) of SIB. Completing a trial-based FA would likely improve safety given a decrease in the possibility of significant harm as the trials were brief and would end after the first occurrence of SIB. A termination criterion was discussed with Farah’s caregiver and included SIB to the point of drawing blood; however, this criterion never occurred at any time.

Four 2-min synthesized test conditions had a matched 2-min control condition (eight conditions in total), which are described below. Farah was shown one of eight colored glowsticks at the start of each condition to enhance discrimination between conditions. Conditions were randomized using an online random number generator. The matched control condition always occurred before the test condition [28]. Each condition was terminated after 2 min elapsed or after the first occurrence of SIB (either a single response or a rapid burst of responses) and its respective consequence. If SIB occurred during the control condition, no consequences were provided; however, the respective test condition then immediately began. For every condition, the tasks (puzzle, stacking Legos, lacing beads), the form of attention (positive physical attention, e.g., a hug or picking Farah up, and positive statements such as “thank you for asking for a hug” or “it’s okay”), and the preferred item (bottle with milk) remained consistent. Each condition was conducted three times. 

Escape from demands to access attention. The control condition entailed providing noncontingent positive, physical attention, and no tasks. During the respective test condition, the clinician presented tasks, and contingent upon SIB, the clinician removed the task and made a positive or comforting statement paired with positive, physical attention for 30 s. If Farah did not respond to tasks, clinicians used three-step prompting (i.e., vocal, model, and physical prompts) to gain compliance.

Access to tangibles and attention. The control condition included providing noncontingent positive, physical attention and access to her bottle. During the matched test condition, Farah was provided access to her bottle and physical attention for 30 s. Once the trial began, access to her bottle and the clinician’s attention were removed. The bottle and physical and comforting attention were provided for 30 s contingent upon SIB. 

Escape from demands to access tangibles. The control condition consisted of noncontingent access to her bottle only and no tasks were delivered. Before the matched test condition, Farah was offered access to her bottle for 30 s, but no attention was provided. Once the trial began, the bottle was removed but remained in sight, and tasks were presented. All other procedures were identical to the escape-from-demands-to-access-attention condition, except the bottle was provided instead of attention contingent on SIB. 

Escape from demands to access attention and tangibles. The control condition was the same as the previously described control conditions with both physical attention and the bottle provided noncontingently. The respective test condition was similar to the previous test conditions including pre-access exposure to both putative reinforcers and delivery of both reinforcers contingent upon SIB. 

## 5. Procedure—Phase 2

### 5.1. Competing Stimulus Assessment (CSA)

The results of the synthesized trial-based FA suggested that Farah’s SIB was maintained by a combination of positive, physical attention, escape from demands, and access to tangibles. However, Farah’s caregiver described how difficult it was to complete tasks or attend to her other child given that Farah would typically engage in SIB almost immediately once attention was removed. As such, clinicians and the caregiver completed a CSA to identify items that might compete with the reinforcer (i.e., positive physical attention) maintaining SIB [38]. Although CSAs are most commonly implemented for automatically reinforced behavior [38], competing items can also be used when attention is the reinforcer for challenging behavior. For example, ref. [39] identified attention as the maintaining variable for destructive behavior for four individuals. Competing stimuli (e.g., headphones) were identified, which successfully decreased destructive behavior when attention was not available, and extinction was in place. CSA items were selected based on caregiver nomination, ease of providing the item, and preference assessment results. 

Clinicians initially conducted the CSA sessions and then taught Farah’s caregiver the procedures. For each session, one tangible item was presented at a time for 2 min. The adult would begin by presenting the item with instructions such as, “I’m going to be busy, so you can play with this toy”. The adult would then remove attention and engage with their phone or book. No consequences were provided following SIB or any other behaviors. To further expedite the assessment process and minimize further opportunities that Farah could engage in SIB, we used the previously conducted standard FA alone test condition data (in which no items were present) as the control condition of the CSA, as has been conducted in past research (c.f. [39]) Each item was assessed three times with each adult. 

### 5.2. Social Validity

Farah’s mother observed all sessions and completed an adapted Treatment Acceptability Rating Form-Revised (TARF-R; [40]) for each synthesized FA condition. The TARF-R includes modified questions to measure the social acceptability of the FA process for each control and test condition [40]. For example, the TARF-R asked questions such as, “I find this approach to be an effective way of assessing my child’s challenging behavior” or “I believe my child experienced discomfort during the assessment”. The TARF-R includes nine questions, with answers based on a 5-point Likert scale, with 1 indicating strong disagreement, to 5 suggesting strong agreement with the statement.

## 6. Results 

The results of the standard FA are depicted in Figure 1. The rates of SIB were undifferentiated across conditions. Although the toy play condition was meant to serve as a control condition (establishing operations absent), high rates of responding were observed in the first two sessions, despite Farah having noncontingent access to preferred items and attention, with no demands. Interestingly, the lowest rates of SIB were noted in the attention conditions. 

The synthesized trial-based FA results in Figure 2 show elevated differentiation in the rates of SIB for three of the four test conditions: escape from demands/attention (*M* = 66%), attention/tangible (*M* = 33%), and escape from demands/attention/tangible (*M* = 66%) compared to their respective control conditions (*M* = 0%). This finding suggests that the function(s) of SIB is likely a combination of these synthesized variables. 

The results from the CSA (Figure 3) identified at least two items that promoted lower SIB, with at least one person, compared to the alone FA control condition (*M* = 0.4 responses per minute). Specifically, the pin toy promoted low rates of SIB with the clinician (*M* = 0.33 responses per minute) or caregiver (*M* = 0), and the music toy produced zero rates of SIB with the caregiver (*M* = 0); however, the duration of engagement was higher with the pin toy (*M* = 1.38 min) compared to the music toy (*M* = 0.07 min) with the caregiver.

The results of the TARF-R (displayed in Table 1) indicate the high social acceptability of the synthesized FA procedures by the caregiver (*M* = 4.5). The caregiver strongly agreed that the FA was acceptable, liked the procedures, would be willing to use the procedures again in the future, and had an overall positive reaction to the assessment. However, the mother agreed that Farah may have experienced discomfort during the test conditions, but not during the control conditions. She also felt neutral that any of the assessment conditions would result in permanent improvement in her child’s challenging behavior. These findings suggest the overall acceptability of the FA process for identifying a function(s) of SIB, despite the caregiver perceiving some possible discomfort for Farah during the test conditions.

## 7. Discussion

The current study provided a unique demonstration of identifying the function(s) of severe SIB using a synthesized trial-based FA format. Similar to the findings by [28,41], the current results showed differentiation in the levels of SIB for three out of four test conditions compared to their respective control conditions. When the establishing operation for escape was present (i.e., demands presented), challenging behavior consistently occurred (although less than in the control condition for the escape-from-demands-to-access-tangibles test condition). However, other reinforcers including attention and access to tangibles (either in isolation or together) also maintained challenging behavior, so it is possible that responding occurred across all test conditions given that one or all of the relevant establishing operations were always present. This finding aligns with research that discusses the potential for false positives from synthesized contingencies, as it is unclear which establishing operation or combination of establishing operations may be evoking challenging behavior [42]. 

Based on the risk of harm to Farah due to the severity of her SIB, it was essential to identify the function(s) of her SIB to develop a function-based treatment. An initial standard FA had been previously conducted with Farah, resulting in ambiguous outcomes, and required 75 min to complete. Thus, the synthesized trial-based FA was selected based on its efficiency, safety, and practicality in determining functions of behavior [27,28,43]. Given the nature of Farah’s SIB, we aimed to minimize exposure to evocative scenarios by keeping sessions brief and only completing the minimum number of trials necessary to identify a function(s) of behavior or competing stimulus. In total, we completed the trial-based FA in less than 42 min using a response-guided approach. Although trial-based FAs typically determine the number of trials to be completed at the onset of the assessment (i.e., a priori, [27]), doing so may result in an unnecessary number of trials completed to confirm a functional relation (e.g., [44]). When assessing severe behavior, this decision can result in the potential for significant and unnecessary harm to oneself or others, should extended and unnecessary sessions be conducted. As such, this issue speaks to the need to identify efficient yet valid assessment and function identification methods when dealing with challenging behavior. Our assessment results also support and extend the findings by [28] in which they combined the trial-based FA and IISCA formats to produce a brief assessment that identified the function(s) of challenging behavior for three participants. 

The results of the CSA identified two items that promoted low rates of SIB when Farah’s caregiver was present, but attention was removed, as compared to the alone control condition. In other words, the two items may have competed with or decreased the value of attention for Farah. Our findings corroborate the few examples of the effective use of competing items for attention-maintained SIB (for a review, see [38]). Thus, the current findings contribute to the small but growing body of research on the effects of competing stimuli on treating attention-maintained SIB [38,45]. 

We also extended research on maximizing efficiency by incorporating FA data (i.e., alone condition) in the CSA as a no-stimulus control condition [39], which may have minimized further occurrences of SIB. Brevity in behavioral assessment and treatment is generally reported to be valued by caregivers [46], who are often facing a range of challenges including stress [47], financial burden [3], and decreased quality of life [4]. However, despite our assessment successes, some limitations are worth noting. Our main goal was to attempt to determine the function(s) of Farah’s SIB to be able to provide treatment recommendations to caregivers promptly. Although we demonstrated clear differentiation of assessment results within six trials per condition (i.e., three trials of test and three trials of control), we were unable to validate the FA or CSA results by conducting a function-based treatment given that the family relocated. As such, the predictive validity of the current findings cannot be verified. Future research should validate the assessment results such as implementing a function-based intervention with the relevant competing stimuli. Anecdotal reports from the family indicated that they employed the behavioral recommendations of providing the competing items when attention needed to be removed at home and minimizing attention following SIB. However, no formal treatment data were collected. Relatedly, another limitation includes the absence of validating the synthesized trial-based FA results to the standard FA methodology with synthesized test conditions (i.e., concurrent validity) as has been conducted in past research evaluating modified FA formats (c.f. [1]). As such, future research should compare findings from both FA formats to corroborate the utility and validity of the synthesized trial-based FA. However, we agree with [48] that, in some cases, prioritizing assessment efficiency and safety in clinical practice may take precedence over high concurrent validity. Future research could consider incorporating modified assessment methods or continuing to combine existing assessments as an alternative to the standard FA. Another limitation is that we did not collect procedural fidelity data during the FAs or the CSA for the clinicians or the caregiver. Thus, although we calculated the acceptable agreement between independent data collectors, these data do not reflect the accuracy of the procedures implemented. Future research should include procedural fidelity measures during the FA and subsequent assessment procedures to increase the validity of the results. 

Lastly, all three test conditions that delivered physical attention as part of the reinforcing consequences showed some differentiation compared to their respective control conditions where attention was provided noncontingently. Trial-based FA research suggests that responding exclusively in the test condition contributes to a valid demonstration of a functional relation if there is no challenging behavior in the contiguously presented control condition [49]. The degree of vertical separation between test and control conditions in a bar graph reportedly varies [50,51], as well as the number of trials required [27]. Recent studies have demonstrated the accurate identification of a function of challenging behavior in as little as three trials [49,50], but as many as twenty trials [27]. Ref. [51] cautioned that using less than five trials may be problematic; however, more research is needed to compare the level of responding with varying numbers of trials as it relates to the final determination of function(s). When dealing with more severe challenging behavior, assessments that maximize efficiency should arguably be prioritized without compromising analytic effectiveness. Refs. [49,50] advocates for the importance of formative procedures (e.g., data-based decision-making) when conducting the trial-based FA so that both analytic precision and efficiency can be achieved. Following this model, we identified differentiation between three test conditions compared to their respective control conditions after three trials, suggesting a potential function(s). Future research should validate the results of the trial-based FA while evaluating the number of trials required to determine a function, based on the methods described previously (e.g., [49,50]).

## 8. Conclusions

In conclusion, after completing a previously inconclusive standard FA, a synthesized trial-based FA successfully identified the function(s) of severe SIB in a safe but quick manner for a young girl with severe SIB. Specifically, the synthesized trial-based FA found differentiation between three test and control conditions in under 42 min, suggesting the evocative variables of attention and escape (in isolation or combined) likely functioned as a reinforcer(s) for SIB. Our findings suggest that a trial-based synthesized FA can be an efficient method to assess challenging behavior when resources or safety are a concern. Based on the barriers of the standard FA [16,52], identifying safe and efficient assessment methods in practice is of utmost importance, especially concerning severely challenging behavior. Given that practitioners report failure to conduct FAs in practice due to clinical challenges [52], creating efficient FA alternatives may provide an optimal method of doing so. A CSA for attention-maintained SIB identified two items that promoted low or zero rates of SIB when attention was removed with at least one adult. Although a formal treatment evaluation was not completed due to the family relocating, treatment recommendations were provided related to using the competing items when attention needed to be removed or withheld. Moreover, to keep treatment recommendations feasible, we provided mainly antecedent recommendations based on the CSA results to minimize the burden and treatment complexity for Farah’s caregivers at home. Future research is necessary to provide additional demonstrations of expedient assessment procedures with formal treatment evaluations to further validate the assessment results. 

## Figures and Tables

**Figure 1 behavsci-14-00372-f001:**
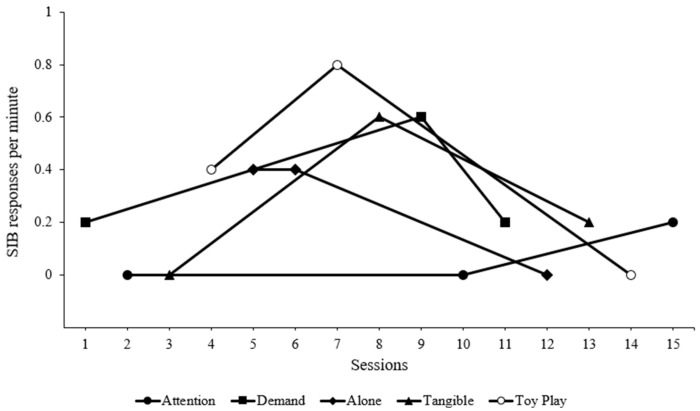
Standard functional analysis results.

**Figure 2 behavsci-14-00372-f002:**
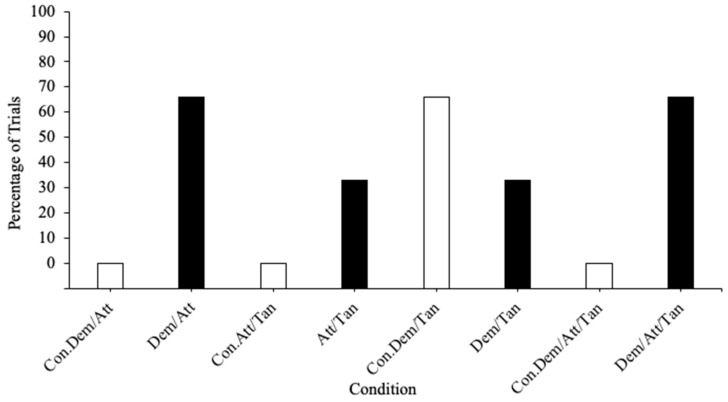
Trial-based synthesized functional analysis results. Note: Con. = control condition; Dem = demand; Att = attention; Tan = tangible. White bars indicate control condition sessions; black bars indicate test condition sessions.

**Figure 3 behavsci-14-00372-f003:**
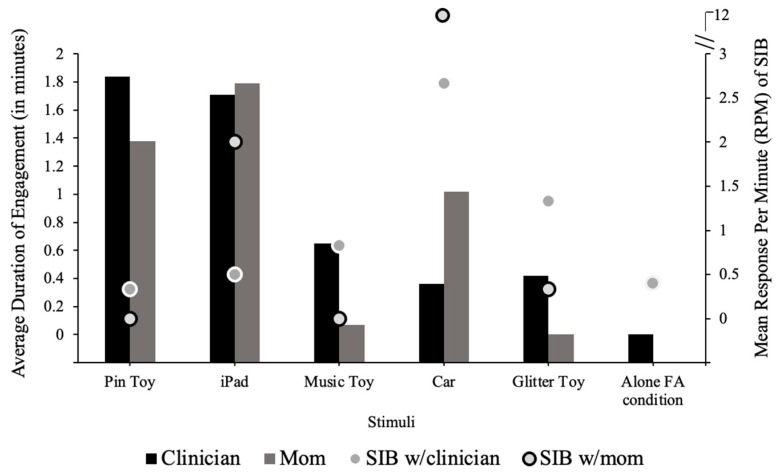
Competing stimulus assessment results.

**Table 1 behavsci-14-00372-t001:** Adapted Treatment Acceptability Rating Form-Revised [40] assessing caregiver acceptability of FA conditions.

Treatment Acceptability Rating Form-Revised (TARF-R)					
Questions	Conditions				
	Controls	Demand/Attention	Demand/tangible	Attention/Tangible	Demand/Attention/Tangible
I find this approach to be an acceptable way of assessing my child’s challenging behavior.	5	5	5	5	5
I would be willing for this procedure to be used again to assess my child’s challenging behavior.	5	5	5	5	5
I believe it would be acceptable to use this assessment without my child’s consent.	5	5	5	5	5
I like the procedures used in this assessment.	5	5	5	5	5
I believe this assessment is likely to be effective in identifying the factors that cause my child’s challenging behavior.	5	5	5	5	5
* I believe that my child experienced discomfort during the assessment.	1	4	4	4	4
I believe the assessment is likely to result in permanent improvement in my child’s challenging behavior.	3	3	3	3	3
I believe it would be acceptable to use this assessment with people who cannot choose assessments for themselves.	5	5	5	5	5
Overall, I had a positive reaction to this assessment.	5	5	5	5	5
Mean Rating	4.8	4.4	4.4	4.4	4.4

Note: The TARF is rated with a 5-point Likert scale, where 1 is strongly disagree and 5 is strongly agree. * This item is negatively worded so that the item’s scores are inversed when calculating the mean rating.

## Data Availability

Functional analysis data are available upon request.

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
