# Peer review of "Toward Maximizing Assessment Efficiency: A Synthesized Trial-Based Functional Analysis and Competing Stimulus Assessment"

_behavsci, 2024, doi:10.3390/bs14050372_

Round 1

Reviewer 1 Report

Comments and Suggestions for Authors

Summary: This n=1 study describes, after a brief but to the point theoretical introduction, a brief trial-based FA using ecologically relevant synthesized contingencies in a 3 year old girl, performed in narrow co creation with the mother of this child, and step by step described and evaluated. The main question of this study is if a faster way of finding relevant stimuli would be possible using a trial-based design. 

Strengths: This article presents a well described exploration and demonstration of an efficient behavioral therapy design that is very usable in clinical practice, with adaption also in adult patients. The line of reasoning is clear, and enough detail has been included. I adresses a clinically relevant question, especially in case of self-mutilation, where some speed in this process can be needed. The conclusions are consistent with the results and sufficiently linked to relevant literature.

Weaknesses: no weakness detected, especially no methodological adaptations or clarifications. The importance of this n=1 design to further development of trial-based FA as a relevant clinical option could be more strongly pronounced, for example by providing a table of weaknesses and strengths of this design. 

Reviewer 2 Report

Comments and Suggestions for Authors

The research questions or purposes need to be stated explicitly. The literature review should be used to motivate the specific research questions rather than providing only the contexts for the case. It is not possible to judge the quality of this study or the manuscript without a clearly formulated research question. Also, it is uncertain if the unique features described in this case are indeed exceptional and different than what has been found in the existing literature.

Comments on the Quality of English Language

Minor errors in references

Reviewer 3 Report

Comments and Suggestions for Authors

I would like to thank you for the opportunity to review the manuscript submitted for consideration in the journal Behavorial Sciences. It is a case report of a girl with ASD. The manuscript is interesting for the field and is well thought out. Below, you will find the considerations for each section.

Title and Abstract Review:

The title is very generalist, it would be appropriate to be able to infer what the manuscript is going to be about, at least some clue to make the content more concrete. The abstract explains the content quite well, however, I would expand a little more on the methodological part and the main results of the case study. Likewise, no research objectives or hypotheses are detected.

Introduction and literature review

In this section, the research is adequately based on the previous one. The epistemological foundation of the research is presented, so its referents are solid. However, no purpose is found in the research beyond showing the case. No objective or hypothesis is presented. It seems that the case study is only carried out to report the facts that occur, without any purpose. If this is not detected in the first sections, the reader does not understand the purpose of the research.

Materials and methods

It is sufficiently developed. Concise and detailed information on the procedure is given. However, the procedure should be included in the methodology, not as a separate item.

Results and discussion.

The results and discussion are poorly organized. There is no clarity in their reading. Mixing these two sections makes them difficult to read because they are not structured.

Conclusions

This section does not exist, although there is a paragraph dedicated to it. Conclusions should be presented as a strong point of the research, as well as the limitations of the research and future lines of research. It is doubtful that the study will contribute relevant novelties to scientific knowledge. The specificity of the study could be of interest if it contributed some innovative component in the instructional design, but this is not the case. It would have to be argued/justified or clarified.

In line with the above, the conclusions are very generic and do not provide new insights into the phenomenon studied. It would be better to clarify which studies are confirmed or refuted by the research carried out, and which studies are complementary. The citations are also not sufficiently clear. In short, an effort should be made to emphasize the value of the conclusions and the novel contribution to knowledge.

Reviewer 4 Report

Comments and Suggestions for Authors

The article "Toward Maximizing Efficiency in Assessment Time for a Young Child with Severe Self-Injurious Behavior" focuses on a novel assessment approach for analyzing a young child with severe self-injurious behavior (SIB). It delves into the application of Synthetic Trial-Based Functional Analysis (FA) in handling cases of SIB and conducts a thorough evaluation of its effectiveness.

After a comprehensive review, the following suggestions for refinement are proposed:

- The article should more emphatically highlight the specific advantages of Synthetic Trial-Based FA over traditional FA, elaborating on how previous studies have attempted to address the limitations of conventional FA, especially in cases of children with severe SIB.

- Detailed descriptions of the participant's background enhance the understanding of the research context. It's recommended to include more information about the criteria for participant selection and the suitability of this specific case for the study.

- Additional details regarding the selection of specific materials and settings, and their alignment with the study's objectives, should be provided.

- Clarification on the specific methods used for data collection and analysis, particularly concerning the accuracy and reliability of the data, would be beneficial.

- Despite providing detailed Interobserver Agreement (IOA) data, further discussion on how this data influences the study results and interpretation of any discrepancies is needed.

- An explanation of why certain test conditions were chosen and how they represent the child's home environment would strengthen the study.

- More detailed reasoning for the selection of specific experimental settings and conditions, and their correlation with the study's hypotheses and objectives, is suggested.

- In describing the CSA method and results, further elucidation of how these results contribute to understanding and treating SIB, as well as their comparison with existing research, is advised.

- An addition of a comparison with existing literature and the potential impact of these findings on future research is needed.

- In discussing the effectiveness of the research results and implementation methods, a deeper exploration of how these methods can be improved or optimized for assessing similar cases in the future would be insightful.

- Although key findings are provided, a more detailed explanation of how these results affect our understanding of SIB and its treatment is necessary.

- The discussion should delve deeper into the impact of these findings on future research and practice, including a broader comparison with previous studies and how this research advances the field.

Reviewer 5 Report

Comments and Suggestions for Authors

As an assessment of the approach taken in this manuscript, I find that the writing is quite comprehensive and flows smoothly. The research also holds significant value, and I recommend publication after minor revisions. In the revision process, the author should primarily focus on two aspects.

Firstly, citations should not appear in the abstract of an academic paper. The abstract should succinctly describe the research methods, results, and insights. The current abstract needs to be rewritten accordingly.

Secondly, the theoretical value of the study should be explicitly addressed. A dedicated explanation of the theoretical significance of the research is necessary.

Round 2

Reviewer 2 Report

Comments and Suggestions for Authors

N/A

Author Response

Thank you for the feedback! We have thoroughly read through the MS in its entirety two times to edit and make changes throughout. We have modified grammar and made statements more concise as needed. 

Reviewer 3 Report

Comments and Suggestions for Authors

The authors have made the necessary corrections, the article can be published under the current conditions.

Author Response

Thank you for the feedback! We have thoroughly read through the MS in its entirety two times to edit and make changes throughout. We have modified grammar and made statements more concise as needed. We hope this  continues to improve the quality of the manuscript.

Reviewer 4 Report

Comments and Suggestions for Authors

The author has addressed most of my concerns. The writing and formatting of the revised manuscript also require careful proofreading.

Author Response

Thank you for the feedback! We have thoroughly read through the MS in its entirety two times to edit and make changes throughout. We have modified grammar and made statements more concise as needed. We hope these changes will contribute to the improvement of the manuscript.